# Tailoring the energy landscape of a bloch point domain wall with curvature

Sandra Ruiz-Gómez [1,2] ✉, Claas Abert [3,4], Pamela Morales-Fernández [1], Claudia Fernández-González [1,2], Sabri Koraltan [3,4,5], Lukas Danesi[3,4], Dieter Suess [3,4], María Varela [6], Gabriel Sánchez-Santolino [6], Núria Bagués[2], Michael Foerster [2], Miguel Ángel Niño [2], Anna Mandziak[7], Dorota Wilgocka-Ślęzak[8], Pawel Nita[7,9], Markus Koenig[1], Sebastian Seifert[1], Aurelio Hierro-Rodriguez [10,11,12], Amalio Fernández-Pacheco [13] & Claire Donnelly [1,14] ✉

Topological defects, or singularities, play a key role in the statics and dynamics of complex systems. In magnetism, Bloch point singularities represent point defects that mediate the nucleation of textures such as skyrmions and hopfions. While these textures are typically stabilised in chiral magnets, the influence of chirality and symmetry breaking on Bloch point singularities remains relatively unexplored. Here, we harness advanced three-dimensional nanofabrication to explore the influence of symmetry breaking on Bloch point textures by introducing controlled nano-curvature in a ferromagnetic nano-wire. Combining X-ray magnetic microscopy with the application of in situ magnetic fields, we demonstrate that Bloch point singularity-containing domain walls are stabilised in straight regions of the sample, and determine that curvature can be used to tune the energy landscape of the Bloch points. Not only are we able to pattern pinning points but, by controlling the gradient of curvature, we define asymmetric potential wells to realise a robust Bloch point texture shift-register with non-reciprocal behaviour. These insights into the influence of symmetry on singularities offer a route to the controlled nucleation and propagation of topological textures, providing opportunities for logic and computing devices.

Topology and topological textures, which mathematically describe fundamental properties of multidimensional objects, have made it possible to compare fields from gravitation[1,2] to liquid crystals[3,4] and magnetism[5,6], providing a simplified view to understand and compare the statics and dynamics of complex systems. One of the key reasons why topology has generated interest is due to the prospect of topological stability, where essential system characteristics persist through smooth transformations[7,8]. To facilitate the transformation of a

[1]Max Planck Institute for Chemical Physics of Solids, Dresden, Germany. [2]ALBA Synchrotron Light Source, CELLS, Cerdanyola del Valles, Barcelona, Spain. [3]Faculty of Physics, University of Vienna, Vienna, Austria. [4]Research Platform MMM Mathematics-Magnetism-Materials, University of Vienna, Vienna, Austria. [5]Vienna Doctoral School in Physics, University of Vienna, Vienna, Austria. [6]Departamento de Física de Materiales & Instituto Pluridisciplinar, Universidad Complutense de Madrid, Madrid, Spain. [7]SOLARIS Synchrotron light Sources, Crakow, Poland. [8]Jerzy Haber Institute of Catalysis and Surface Chemistry, PAC, Krakow, Poland. [9]Faculty of Physics, Astronomy and Applied Computer Science, Jagiellonian University, Crakow, Poland. [10]Depto. Física, Universidad de Oviedo, Oviedo, Spain. [11]CINN (CSIC-Universidad de Oviedo), El Entrego, Spain. [12]SUPA School of Physics and Astronomy, University of Glasgow, Glasgow, UK. [13]Institute of Applied Physics, TU Wien, Vienna, Austria. [14]International Institute for Sustainability with Knotted Chiral Meta Matter (WPI-SKCM2), Hiroshima University, Hiroshima, Japan. ✉e-mail: srgomez@ucm.es; claire.donnelly@cpfs.mpg.de

system's topology, the introduction of defects, which disrupt the assumption of a continuous unit vector field, is required[9-12]. In three dimensions, these defects take the form of point defects that have been observed across diverse domains, including gravitational fields (black holes)[13], confined liquid crystals (commonly known as hedgehogs)[14,15] and magnetism (Bloch points)[16,17]. Despite their significant role in state transformations, our understanding of these point defects remains limited due to the challenges in studying, isolating, and controlling them.

In this context, magnetism offers a unique opportunity to delve into the fundamental properties of such defects, allowing us to stabilize and image point defects known as Bloch point singularities[18-22] where, at length scales below 10 nm, the magnetization vanishes[23]. Bloch points, which form as part of static magnetic configurations such as vortex rings, chiral skyrmion bobbers or Bloch point domain walls (BPDW)[24-26], play a key role in mediating dynamic topological transformations such as the nucleation and evolution of skyrmions and the decay of hopfions in chiral magnets[27]. However, although the vast majority of studies of topological magnetism take place in chiral magnets, it is not yet understood how the chirality or symmetry of a system could be used to stabilize, or indeed control the behavior of these topological Bloch point defects.

To determine the influence of symmetry on Bloch point textures, we require two aspects: First, we need a precise control of the local symmetry that typically is achievable via the Dzyalozinskyi Moriya interaction (DMI) with interface-engineering, thin film or crystal growth. Second, we require the ability to reliably nucleate isolated Bloch points textures, which so far has only been possible in electro-deposited cylindrical nanowires, for which the introduction of symmetry breaking or chirality in the material is limited[28,29]. However, until now combining both aspects has not been possible.

Here, we overcome this limitation by using geometric, i.e. curvature-induced symmetry breaking, which can introduce effective chirality into magnetic systems[30-33]. We locally modify the symmetry in a cylindrical nanowire by introducing curvature, allowing us to explore the stability and develop control of Bloch point textures.

## Results

We fabricated the curved cylindrical nanostructures with advanced 3D nanopatterning. Focused electron beam induced deposition combined with computer-aided design (f3ast software) makes possible the direct fabrication of intricate 3D structures with typical spatial resolutions on the order of a few tens of nanometers (see Fig. 1a)[34,35]. Following this approach, we design and fabricate a nanowire with cylindrical cross-section where we introduce regions with local radii of curvature spanning from 250 nm to 2 μm (Fig. 1b, c), each distinctly colored in Fig. 1d. These regions of curvature are separated by straight segments, colored in gray.

To determine the influence of local symmetry breaking on a Bloch point texture, we nucleate BPDW, and then propagate them through regions of varying curvature, tracking their position. We track the position of the BPDW using shadow X-ray photoemission electron microscopy (X-PEEM), where the shadow of the suspended nanowire is directly imaged in transmission[36]. We combine this imaging mode with the use of magnetic sample holders that allow the in-situ application of magnetic fields. A sketch of the measurement configuration is presented in Fig. 2a[37,38].

The three-dimensional nature of these structures, suspended above the substrate by a supportive leg (as depicted in the left part of Fig. 1c), results in images with full visibility of their shadows. The shadows are elongated by approximately a factor of 3.5 along the beam direction due to the grazing incidence of the X-rays (16°). This provides an effective increase of the spatial resolution of the magnetic state in that direction, making it possible to resolve the structure of domain walls in suspended nanostructures. Figure 2b shows an X-ray

absorption spectroscopy (XAS) image of an undulating structure measured at the Co $L_3$ edge, with the objective lens adjusted to ensure that the shadow of the structure remains in focus. Together with XAS images, dichroic images (XMCD images) can be measured by obtaining the pixel-by-pixel asymmetry between images measured with opposite X-ray helicities. As it can be seen in the sketch of Fig. 2c, such an XMCD image gives a contrast proportional to the magnetization component along the X-ray direction, leading to an alternating XMCD contrast for an undulating structure in a single domain state.

In order to induce the formation of a domain wall within the nanowire, two distinct methods can be employed: applying a magnetic field perpendicular to the long axis of the nanostructure (blue squares, Fig. 2d–f), or parallel to it (pink squares, Fig. 2g–i). In cylindrical nanowires, there are generally these types of domain walls that occur: Transverse domain walls (TDW) and BPDW, also known as Vortex domain walls[12,39-41]. TDWs are characterized by a rotation of the magnetization within the domain wall perpendicular to the long axis of the nanowire[42]. In contrast, BPDWs exhibit a curling of magnetic moments within a plane perpendicular to the wire axis, featuring a Bloch point at their center[26]. So far, the effect of the local symmetry breaking induced by curvature has been considered only in the context of TDWs, which are chiral objects. In this context, the local symmetry breaking can be thought of as introducing chirality into the system, and has been shown to select the chirality of TDWs[43-45].

In our experiments, to distinguish between the types of domain walls that can be generated, we measure XMCD images with X-rays directed perpendicular to the section of the nanowire housing the domain wall[26]. In the first scenario, where the domain walls are initialized by a saturating field perpendicular to the long axis, the domain wall is found in the curved section of the nanowire. Here, only a uniform white contrast is observed, indicating the presence of a transverse-vortex domain wall with its magnetization aligned with the curvature towards the right of the image, as can be seen in the schematics (Fig. 2e, f). This result agrees with previous studies that have shown that curvature affects the energy of TDW, breaking their degeneracy and favoring one rotational direction over the other[31]. In the second case (when the magnetic field is applied parallel to the nanostructure), we observe a black-white contrast pattern, characteristic of the magnetization curling around the Bloch point (as shown in Fig. 2i). We note that, unlike the TDW, the BPDW is never found in the curved region but rather in the straight section between two curved regions. The BPDW remains in this position even under the influence of small magnetic fields of up to 20 mT (see supplementary information), indicating that it is a well-defined local minimum of energy.

To understand the influence of curvature-induced symmetry breaking on the energetics of the BPDW, we perform finite element micromagnetic simulations[46], determining the evolution of the energy of a BPDW as it moves through curved and straight regions of the sample. We simulate a 70 nm diameter cylindrical nanowire, with two regions of different magnitude of curvature, as shown in Fig. 3a. BPDWs were initialized at different positions between the center points of the two curved region (Fig. 3b), and the transition path between the two states was found by applying the string method in a full micromagnetic model[46,47] (see Methods). To distinguish between the contribution of the singularity to the pinning and the surrounding configuration, we repeat the simulation for a nanotube with an inner diameter of 16 nm, which hosts a pure vortex wall without the Bloch point. The energy landscape for the nanotubes is shown in Fig. 3f. In both cases one first observes that as the domain wall propagates from the region of lower curvature (left, I) to the region of higher curvature (right, III) via an effectively straight section (II), the energy of the domain wall drops sharply, indicating that there is an energy minimum corresponding to the domain wall sitting in the straight region of the structure (II). Although the energy landscape of the BPDW simulated in the nanowire (Fig. 3b) shows higher noise caused by the pinning of the

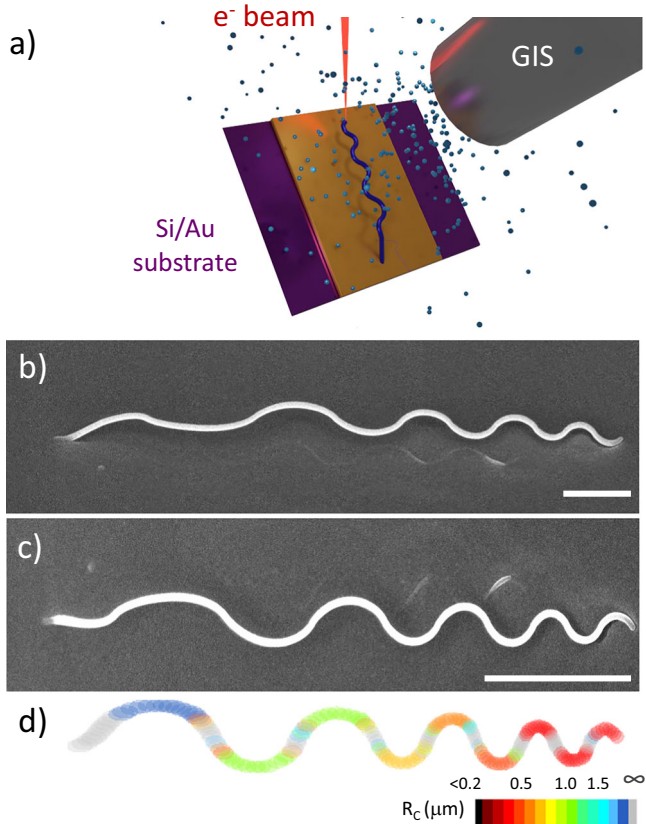

**Fig. 1 | 3D nanoprinting of complex 3D curved nanostructures. a** Schematic of the FEBID process. **b** SEM image of an undulating structure fabricated with FEBID using the $Co_2(CO)_8$ precursor on a Si substrate coated with an Au film. The image has been taken with a tilt of 45°. **c** SEM image from a top view of the same structure and **d** Schematics of the structures in which the color code indicates the radius of curvature along the structure. The scale bar is $1\,\mu m$ in all figures.

BP at the irregular finite element mesh, the energy landscape does not change significantly when transitioning from the wire to the tube (Fig. 3f). This indicates that the curvature effect is dominated by the vortex configuration and not the central singularity. The underlying cause of this energy minimum becomes clear when we consider the curvature-dependence of the exchange and magnetostatic energies separately (shown in SI). The exchange energy of the domain wall in general decreases with increasing curvature, consistent with curvature-induced pinning of transverse walls, due to the reduction in the total rotated angle of the magnetization across the domain wall. In contrast, the magnetostatic energy increases with increasing curvature and, most noticeably, exhibits a sharp drop when the domain wall is in the effectively straight interface region where the curvature is 0 ($\kappa = 0$). Since the total energy is dominated by the magnetostatic energy, the straight interface region represents a local energy minimum, and thus a pinning point for the domain wall.

The curvature-evolution of the magnetostatic energy can be understood considering the local magnetostatic charges of the domain wall. Specifically, the BPDW forms in systems where the magnetostatic energy dominates, to minimize surface charges at the cost of local volume charges and exchange energy in the vicinity of the Bloch point singularity. In straight regions, the symmetry and shape of the domain wall remain unchanged. However, as the domain wall propagates through curved regions, the breaking of symmetry leads to an increase in magnetostatic energy. This increase in magnetostatic energy dominates the overall energy, i.e., in this case, the impact of curvature can be considered a magnetostatic effect. An intuitive picture of this effect can be obtained by considering the symmetry of the

Bloch point domain wall: indeed, the structure is an achiral, symmetric magnetic texture, in contrast to the transverse domain wall, which is intrinsically chiral. Just as a transverse domain wall will have a lower energy in a region of preferred curvature-induced chirality, the energy of the symmetric Bloch point domain wall will be lower in the symmetric, straight region of the structure.

To experimentally confirm the curvature-dependence of the Bloch point domain wall energy, we propagate the domain wall through the nanostructure consisting of regions of alternating, increasing curvature with an external magnetic field. We apply 1 s pulses of magnetic field of increasing magnitude parallel to the long axis of the structure and track the position of the domain wall between pulses at remanence, as shown in Fig. 4a. As we progressively increase the magnitude of the applied magnetic field pulses, the domain wall propagates along the structure in discrete steps, moving from one straight region to the next, as shown in Fig. 4a. In fact, among a total of 29 observations of propagating domain walls, we consistently observed them pinned within the straight regions, corroborating our observation that the Bloch point texture exhibits a preference for symmetric, straight regions (refer to supplementary information for further details and simulations).

As the BPDW propagates along the structure, the degree of curvature in the curved regions increases. This increase in curvature is reflected in the strength of the depinning field, which also increases along the structure length. When we plot the depinning field of the domain wall as a function of radius of curvature in Fig. 4b, we notice two main aspects. First, we observe a non-linear decrease of the depinning field with radius of curvature (Fig. 4b). When we account for the structure's geometry[48], specifically the component of the field perpendicular to the domain wall, we observe that the depinning field increases linearly with the curvature (see Fig. 4d), demonstrating a route to design the energy landscape of the Bloch point domain wall with curvature--induced symmetry breaking. Second, the depinning field of the Bloch point wall does not tend asymptotically to zero as the curvature decreases, as one might expect in a system with low-pinning, rather, it tends to a well-defined offset pinning value (marked with a red dashed line in Fig. 4(b)). For pinning from grain boundaries or defects, the position, and magnitude of the depinning fields typically shows a large distribution[49]. In contrast, here this pinning offset is consistent across different nanostructures, suggesting that this pinning could be due to the textured crystalline structure as observed by aberration-corrected scanning transmission electron microscopy (Supplementary Fig. S10) or it could be due to the nature of the Bloch point wall itself. Indeed, it has been predicted that due to the abrupt nature of a Bloch point singularity, such textures could be susceptible to pinning from the atomic lattice[50,51].

To model the pinning of the Bloch point singularity on the atomic lattice, we perform finite-difference micromagnetic simulations of nanowires containing a BPDW, systematically reducing the cell size from the exchange length (~4 nm) down to below the atomic lattice spacing (<1 nm). In this context, we model the exchange interaction as an atomistic Heisenberg exchange, where atomic sites are placed at the centers of the simulation cells. The exchange integral is scaled with the lattice constant $a$ as $J = A.a$, with $A$ the exchange constant, to ensure consistency with the macroscopic material parameters. This setup allows us to mimic the presence of the atomic lattice and investigate how it can act as a pinning potential for the Bloch point within a continuum micromagnetic framework. Indeed, when we plot the depinning field as a function of cell size in Fig. 4e, we observe a strong influence of the mesh on the pinning of the domain wall, which becomes non-negligible as the cell size approaches the atomic spacing determined with transmission electron microscopy (See Methods/ SI). This indicates that the deterministic pinning observed in the experiments could be related to the pinning of the Bloch point singularity on the atomic lattice within the nanostructures, although the contribution

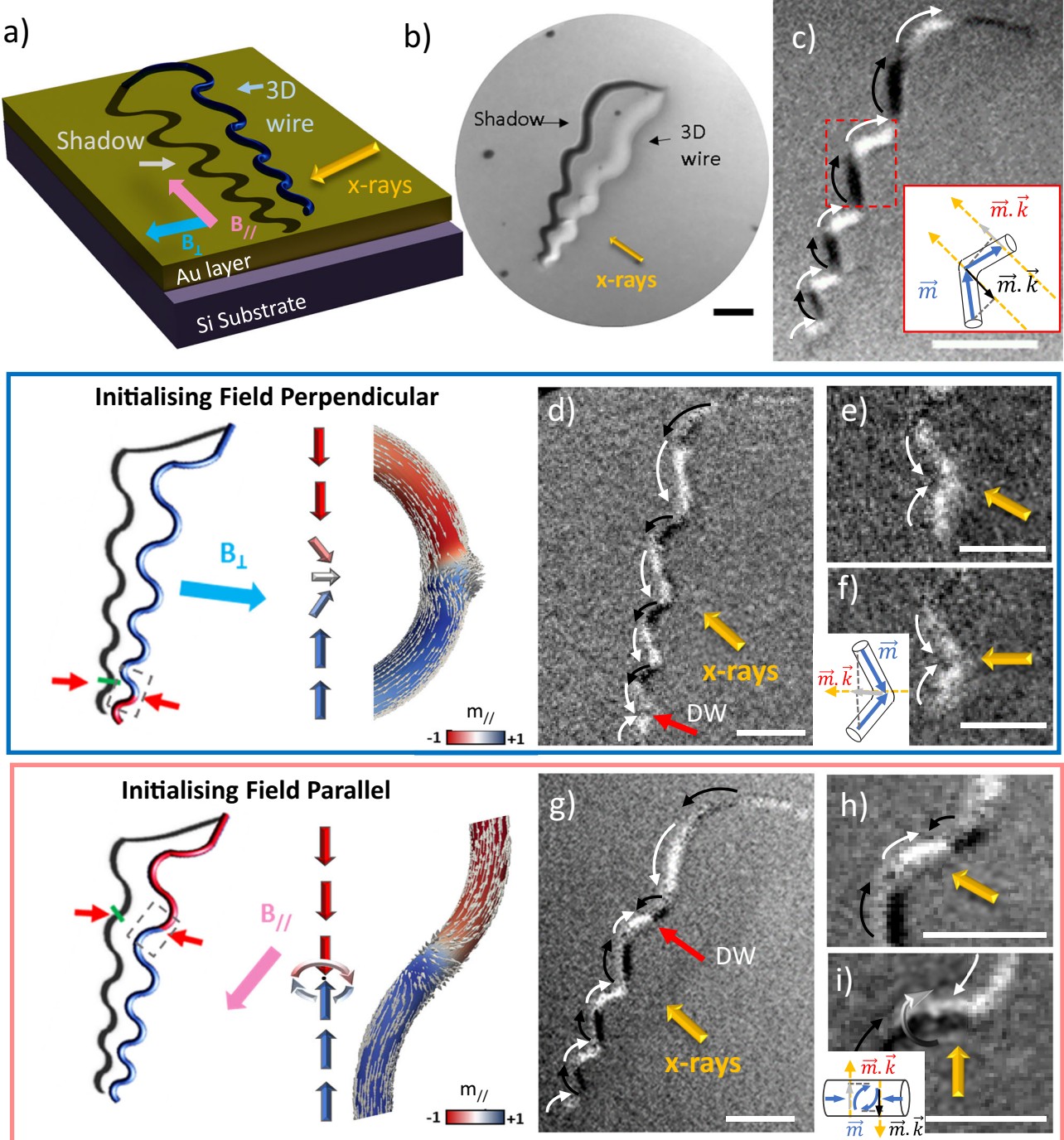

**Fig. 2 | Nucleation of TDW and BPDWs in curved nanostructures imaged by XMCD-PEEM. a** Schematics of the measurement configuration of XMCD-PEEM in shadow mode. The nanostructure is illuminated with circularly polarized X-rays incident at an angle of 16° with the sample surface and the emitted photoelectrons are collected. The nanowire magnetic structure is analyzed by photoelectrons emitted from the substrate by photons which have previously passed through the wire (transmission). **b** XAS image of an undulating structure measured at the Co $L_3$ edge. **c** XMCD image measured at the Co $L_3$ edge for the structure shown in (**b**) in a single domain state with the magnetization pointing down along the structure. The sketch shows the projection of the magnetization with respect to the X-rays direction for the region marked with a red square. White (black) contrast in the XMCD images corresponds to a positive (negative) projection of the magnetization onto the beam direction. **d** XMCD image of the same structure after the nucleation of a domain wall applying magnetic field perpendicular to the long axis of the

nanostructure. The position of domain wall is denoted with a red arrow. **e** Zoom of the previous image in the region where the domain wall is located. **f** Zoom of the same domain wall after rotating the sample (25°) to measure with the X-rays oriented perpendicular to the wire. Yellow arrows indicate the direction of the incoming beam. The sketch shows the projection of the magnetization with respect to the X-ray direction. **g** XMCD image of the same structure after the nucleation of a domain wall applying a magnetic field parallel to the long axis of the nanostructure. The position of the domain wall is denoted with a red arrow. **h** Zoom of the previous image in the region where the domain wall is located. **i** Zoom of the same domain wall after rotating the sample to measure with the X-rays oriented perpendicular to the wire. Yellow arrows indicate the direction of the incoming beam. The sketch shows the projection of the magnetization with respect to the X-ray direction. The scale bar is 1 μm in all cases.

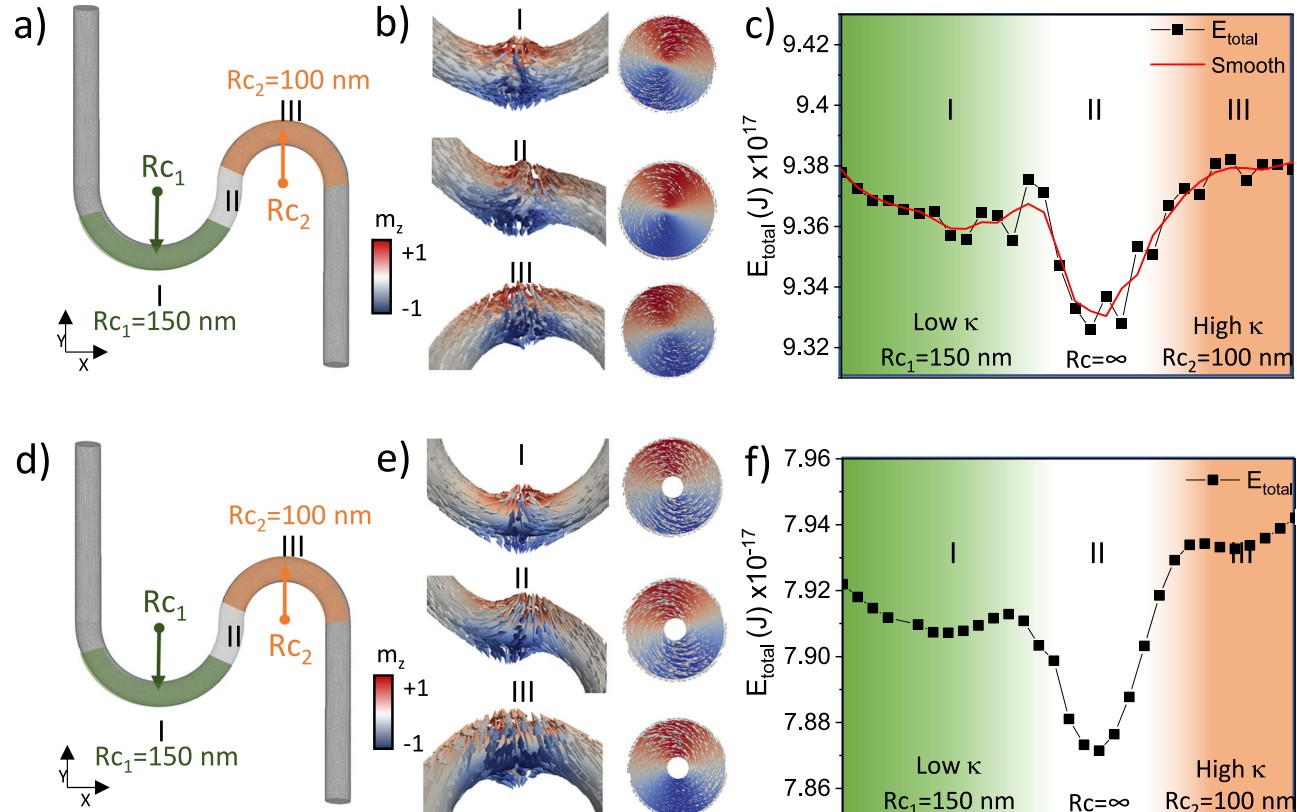

**Fig. 3 | Simulated energy landscape of a BPDW (nanowire) and a VW (nanotube) at curved and straight sections of a nanostructure.** A schematic of the geometry is shown in (**a**), with example positions of BPDWs and their central cross sections shown in (**b**). **c** The energy landscape reveals a well-defined potential minimum in the straight region (II, white), with the total energy of the BPDW increasing in regions of increasing curvature. **d** Schematic of the geometry of the nanotube with example positions of vortex walls and their central cross sections shown in (**e**). **f** The energy landscape reveals a well-defined potential minimum in the straight region (II, white), with the total energy of the vortex wall increasing in regions of increasing curvature.

of other factors related to the geometry and Bloch point spin structure cannot be fully discarded. This pinning of the Bloch point singularity on the atomic lattice is highly relevant not only for the dynamics of BPDW, but also for other configurations and dynamic processes containing Bloch point singularities.

Having determined that the Bloch point singularity strongly influences the behavior of the BPDW, we next consider the effect of curvature on the pinning of the texture. To determine the influence of curvature on this surrounding configuration—and to disentangle the effect of the central Bloch point singularity—we perform separate finite element simulations of a nanowire structure with the central 16 nm containing the singularity removed, effectively a thick-walled nanotube. Here, the geometry is the same as in the previous simulation presented in Fig. 3, and the domain wall is pushed with a magnetic field from a region of lower to higher curvature, via a straight section. When we consider the dependence of the depinning field on the radius of curvature, shown in Fig. 4(g), we observe a close qualitative agreement between the simulations and the experiments, reproducing the effect of curvature on the depinning field. In this way, we attribute the curvature-induced pinning to the surrounding texture of the domain wall. Considering these two contributions together in Fig. 4f, the pinning of the singularity on the atomic lattice, and the influence of curvature on the vortex part of the domain wall, we are able to reproduce the features of the experimental data, confirming the importance of both the central singularity, and the surrounding configuration.

So far, we have explored the unidirectional movement of domain walls, thus probing one side of the potential wells of the Bloch points textures. However, as the local curvature defines the energy barriers in

our system, it is possible for the potential wells to exhibit asymmetry due to neighboring energy barriers of different heights, as predicted by the micromagnetic simulations in Fig. 3. This asymmetry would manifest as a non-reciprocity in the depinning fields of the domain wall, making it easier to propagate in one direction than in the other. We determine the asymmetry of the potential wells at the pinning points in the straight regions of the nanowire by measuring the depinning fields of the BPDW for both positive and negative field directions. Indeed, for a Bloch point domain wall in a straight region (for example straight region #1) situated between two curved regions of varying magnitude (image 2 of Fig. 5a), the propagation field required in one direction (to the right) is higher than in the other (to the left), thereby exposing the inherent asymmetry within the potential well. Repeating this along the nanostructure, we map the energy barriers corresponding to individual curved regions, finding that they are symmetric and that the height of the energy barrier can be set by varying the magnitude of the curvature. Consequently, the energy landscape of the nanostructure is composed of symmetric energy barriers interspersed with asymmetric wells, resulting in an inherently asymmetric propagation pattern of the domain wall, i.e., a magnetic ratchet shift register, entirely determined by the structural geometry, as can be seen in Fig. 5b[52,53].

In conclusion, we have determined the influence of curvature-induced symmetry breaking on the energy landscape of Bloch point textures in a model three-dimensional nanostructure. With advanced 3D nanopatterning, we control the local symmetry breaking by defining curved regions along the nanostructure. In this way, we demonstrate that BPDW predominantly exist in straight, symmetric regions of

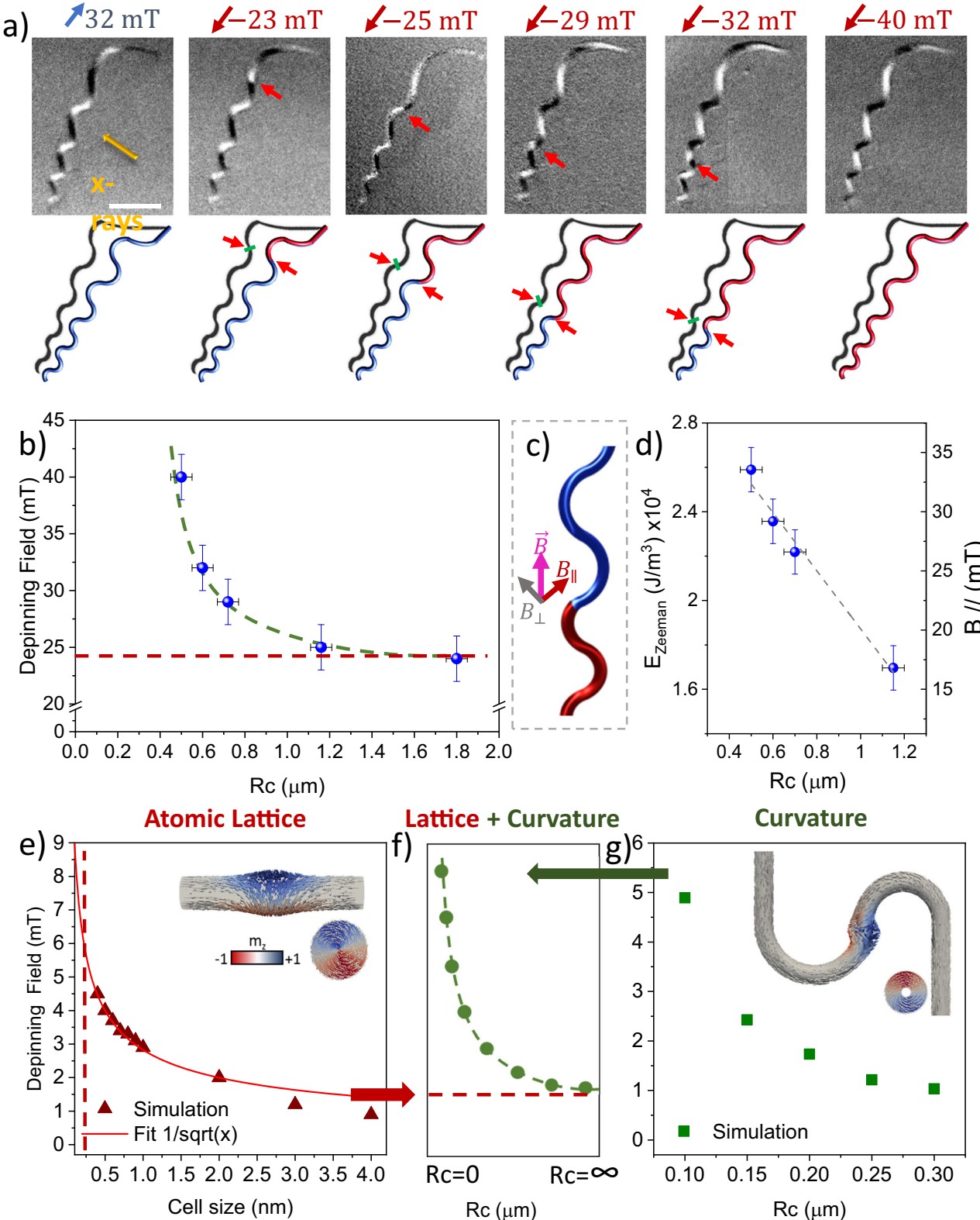

the sample, with curved regions of local symmetry breaking leading to an increase of their energy.

We exploit this finding by introducing regions of varying curvature to control the energy landscape of the Bloch points textures, designing well-defined pinning points in the system. These pinning points are potential wells that exist between two energy barriers, whose height is defined by the local magnitude of curvature. By locally patterning the curvature, we define potential wells that are asymmetric

due to neighboring energy barriers of different heights, which lead to a non-reciprocal motion of the Bloch point textures. We demonstrate this control of the energy landscape by realizing a robust Bloch point shift-register, with tunable depinning fields, and non-reciprocal behavior.

In this work, by micromagnetic simulations, we are able to disentangle the influence of the singularity and the longer lengthscale magnetic configuration of the Bloch point domain wall, determining

**Fig. 4 | Experimental demonstration of curvature induced pinning of a BPDW.** **a** Magnetic configuration of the undulating structure after the application of magnetic fields. The upper row shows the XMCD images measured at the Co $L_3$ edge for the same structure before saturation (I) and after applying a magnetic field sequence of −24 mT (II), −25 mT (III) −29 mT (IV), −32 mT (V) and −40 mT (VI). The XMCD signal is coming from the shadow of the structure where black and white contrast means a component of the magnetization parallel or antiparallel to the X-rays direction. The lower row shows schematics of the magnetization of the structure for each XMCD image where red indicates magnetization pointing up and blue corresponds to magnetization pointing down. The scale bar is $1\,\mu$m in all images. **b** Depinning field as a function of radius of curvature extracted from a

sequence of experiments as the one shown in (**a**). **c** Sketch of the component of the magnetic fields with respect to the domain wall. **d** Component of the magnetic field parallel to the wire as a function of the radius of curvature and Zeeman energy calculated from the depinning field shown in panel b as a function of the radius of curvature. **e** Depinning field as a function of the cell size for a nanowire of 70 nm of diameter containing a BPDW. **g** Depinning field as a function of the radius of curvature for a simulated nanotube of 70 nm of diameter containing a vortex domain wall using a cell size of 4 nm. **f** The expected pinning of a BPDW as a function of curvature arising from the contributions of the atomic lattice (**e**) and the curvature (**g**), which qualitatively reproduces the experimental behavior in (**b**).

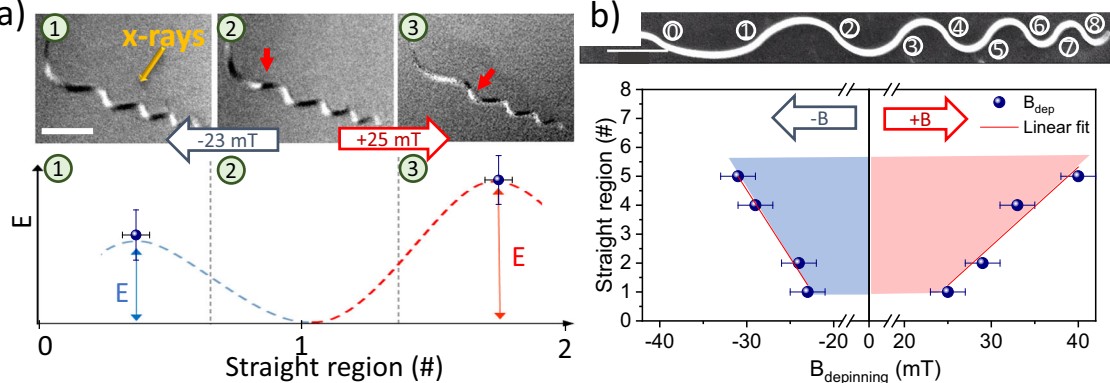

**Fig. 5 | Non-reciprocal motion of BPDWs arising from symmetric potential barriers and asymmetric potential wells in a curved nanostructure.** **a** Top panel: XMCD images of the structure, identifying a domain wall at a straight pinning point #1 (2), and the propagation to the left (1) and to the right (3) after the application of field pulses of varying magnitude. Bottom panel: Schematic of the Energy profile of the potential well of the BPDW in the straight region extracted from the depinning fields, showing an asymmetric potential well corresponding to positive (blue

dashed line) and negative (red dashed line) domain wall propagation. **b** SEM image of the structure denotating the straight regions with numbers. In the bottom panel the depinning field needed to propagate the domain wall through the right (positive fields) and to the left (negative field) is represented together with linear fits (red line). Solid circles represent experimentally determined values. See SI for full dataset.

that both impact the behavior of the texture. The insight into the relationship between Bloch point textures and the symmetry of a system offers an important insight into the contribution to the energy of these elusive singularities, and a route to control them. Indeed, Bloch point singularities play an important role in the formation of topological textures: with Bloch point-containing chiral bobbers mediating the nucleation of skyrmion tubes from surfaces, and Bloch point-containing magnetic torons predicted to be involved in the formation of magnetic hopfions[54]. As such textures are observed in chiral magnets[55,56], an understanding of the influence of symmetry and chirality on Bloch point textures is key to controlling such nucleation processes. Our experimental demonstration that Bloch point textures are stable in symmetric straight regions that are effectively achiral, or interfaces between two regions with symmetry breaking, highlights a future direction for the controlled nucleation of complex 3D textures in chiral magnets.

This concept of harnessing patterned curvature to experimentally explore the influence of local symmetry breaking is not limited to static magnetic configurations: we envisage that this curvature-induced chirality tuning could be used to probe the influence of chirality on magnetization dynamics, such as magnonics, with potential applications beyond magnetism. Indeed, there have been first theoretical predictions that curvature-induced symmetry breaking could lead to tunable chirality in superconductors[57] and superconductor-magnet heterostructures[58], as well as for van der Waal magnets[59]. We envisage that curvature-induced tuning will not only provide key insights into the physics of broader quantum materials, but will also provide a route to tune and control the emergent behavior for future applications. This control, in combination with more complex three-dimensional

architectures, will provide new opportunities for high density and interconnected logic devices[60,61].

## Methods
### Fabrication
The 3D cobalt undulating nanostructures were grown using focused electron beam deposition (FEBID) with the Fei Helios G3 Ga-FIB system at the Max Planck Institute for Chemical Physics of Solids in Dresden. The structures were fabricated on top of Si substrates with a 15 nm thick Au layer to improve the conductivity. The structures were designed using FreeCAD software and converted to beam scanning patterns using the f3ast software[34]. The parameters chose for the growth of the structures were an acceleration voltage of 6 kV and a current of 43 pA. For these parameters, the growth times varied from 10 to 20 min. The radius of curvatures along the structures were measured with a Scanning Electron Microscopy (SEM). After deposition, the samples were annealed for 30min at 250 °C in ultra-high vacuum conditions to avoid the deformation of the structures during XMCD-PEEM measurements.

### Shadow-XPEEM
Shadow XMCD-PEEM measurements were performed at CIRCE beamline[37,38] at Alba synchrotron and at DEMETER beamline at SOLARIS synchrotron. The structures were imaged at the Fe $L_3$-edge measuring images with opposite photon helicities, and subtracted pixel by pixel to determine the in-plane magnetization component along the X-ray direction with nanometer resolution. In-plane magnetic fields were applied in situ using the uniaxial sample holder available at the CIRCE beamline. The sample holder provides a

maximum in-plane magnetic field of 80mT/A. All the XMCD-PEEM images were acquired in remanence after the application of magnetic fields.

## Micromagnetic simulations

Micromagnetic simulations were performed using the finite-element method. In order to investigate the energy of the Bloch-point domain wall depending on the curvature of the sample, a nanotube with a diameter of 70 nm and an inner diameter of 16 nm comprised of two 180 degree arcs with radii 200 nm and 150 nm was considered. This structure was extended by 500 nm long straight sections in both directions to minimize the influence of the magnetic surface charges at the ends of the wire onto the domain- wall energy, see supplementary. An initial transition path for the domain-wall moving from the center of the first arc, to the center of the second arc, was initialized by parametrizing a Bloch- point domain wall at 40 sample points. This initial transition path was numerically evolved employing the string method. In each iteration of the string method, every magnetization configuration in the transition path is driven a certain amount towards the energetic equilibrium using the steepest descent method. The resulting magnetization configurations are then rearranged on the transition path in an equidistant fashion using cubic interpolation.

The energies of the magnetization configurations in the converged transition path are shown in Figure S4 of the supplementary. Since the energy of a Bloch-point domain wall is dominated by the exchange contribution of the Bloch point, the numerical results are subject to significant noise caused by the irregular cell sizes in the tetragonal finite-element mesh despite the choice of 4 nm as a mesh cell size. This pinning of the Bloch point on the mesh is confirmed by simulating the pinning field of a BPDW in a straight nanowire with 70 nm diameter and a length of 250 nm using the finite-difference code magnum.np[62]. To this end, we stabilize the BPDW in the center of the wire and compensate for the surface charges at the ends of the wire in order to eliminate finite-size effects. We then perform a hysteresis simulation with different mesh sizes to determine the dependence of the depinning field from the discretization. The results shown in Fig. 4(e) indicate that the pinning field increases with decreasing simulation cell size, which can be accounted to the exchange energy of the bloch point. A quantitative description of this pinning process is expected when choosing the simulation cell size equal to the lattice constant of the material since the micromagnetic modeling of the exchange interaction converges to the Heisenberg model in this case. While we were not able to scale the mesh size down to the lattice constant due to memory limitations, the trend of the depinning-field dependence on the mesh size indicates a good agreement with the experimental findings where a constant value of the depinning field s obtained for large radius of curvature.

## STEM measurements

Aberration-corrected scanning transmission electron microscopy STEM combined with electron energy-loss spectroscopy (EELS) measurements were performed using a spherical aberration-corrected JEOL ARM200cF operated at 200 kV and equipped with a Gatan Quantum EEL spectrometer. The zero-loss peak was used to correct any energy shift of core-loss signals. Prior to generating elemental signal maps, EE spectra were de-noised using multivariate statistical analysis (MSA) available in Gatan Microscopy Suite (GMS) software.

## Data availability

All data associated with this manuscript are available open access at https://doi.org/10.5281/zenodo.15209659.

## Code availability

All analysis code associated with the work of this manuscript are available open access at https://doi.org/10.5281/zenodo.15209659.

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

## Acknowledgements

These experiments were performed at CIRCE beamline at ALBA Synchrotron Light Facility and at DEMETER beamline at National Synchrotron Radiation Center SOLARIS with the collaboration of ALBA and SOLARIS staff. S. R-G, P. M-F, C. F-G and C.D. acknowledge funding from the Max Planck Society Lise Meitner Excellence Program and funding from the European Research Council (ERC) under the ERC Starting Grant No. 3DNANOQUANT 101116043. P.F-G acknowledges support from the International Max Planck Research School for Chemistry and Physics of Quantum Materials. S. R-G acknowledges support from the Humboldt foundation grant 1223621 and Marie Curie fellowship grant GAP-101061612. C.A. acknowledges funding from FWF through the projects. MF and MAN acknowledge funding through MICIN, project PID2021-122980OB-C54. N.B and S. R.-G. acknowledge funding through Advanced Materials program supported by MCIN with funding from the European Union NextGenerationEU (PRTR-C17.I1) and by the Generalitat de Catalunya. A. F.-P. Acknowledges funding from the European Community under the Horizon 2020 program. Contract NO 101001290 (3DNANOMAG). G.S.-S. acknowledges financial support from Grant RYC2022-038027-I funded by MICIU/AEI/10.13039/501100011033 and by ESF + . Electron microscopy observations at ICTS ELECMI (Centro Nacional de Microscopia Electrónica) sponsored by grants Spanish MICINN PID2021-122980OB-C51 and Comunidad de Madrid "Materiales avanzados" MAD2D-UCM3.

## Author contributions

C.D., A.F.-P., and S.R.-G. conceived the project. The samples were fabricated by S.R.-G. with the support of A.F.-P., C.D., M.K. and S.S. The synchrotron measurements were performed by S.R.-G., C.D., P.M.-F. and C.F.-G. with support of M.F., M.-A.N., D.W.-S. and A.M., P.N., and data analysis by S.R.-G. and C.D. STEM measurement were performed by M. V., G. S.-S., N. B. and S. R.-G. Micromagnetic simulations were performed by C.A., L.D., S. K., C.D. and S.R.-G. with the support of D.S. The results have been interpreted by S.R.-G., C.D., A.H.-R., A.F.-P., C.A. and S.K. S.R.-G. and C.D. wrote the manuscript with contributions from all authors.

## Funding

## Competing interests

The authors declare no competing interests.
