## [Peer Review File · Nature Communications]

Tailoring the energy landscape of a Bloch point domain wall with curvature

Corresponding Author: Dr Sandra Ruiz-Gómez

Version 0:

Reviewer comments:

Reviewer #1

(Remarks to the Author)

Authors provide sufficient explanation on demand of the Referee. The revised manuscript is properly improved to deserve publication in Nature Communications.

Reviewer #2

(Remarks to the Author)

In this study, domain wall (DW) motion in a curved 3D nanostructure was investigated. The authors have provided responses to the previous round of review comments. The microstructural characterization is particularly important for identifying the mechanisms underlying DW movement. The manuscript has been much improved. However, I still have the following concerns.

1) In many places of the manuscript, the effect of chirality was mentioned. For example, the authors state in Line 284 that they control local symmetry breaking and effective DMI by defining curved regions along the nanostructure. However, the observed pinning appears to result primarily from variations in magnetostatic energy due to changes in the radius of curvature, rather than from the chirality of the curvature. This is also evident from their micromagnetic simulations of the vortex domain wall.

2) The authors report that reducing the mesh size affects the pinning energy of the Bloch point domain wall (BPDW) as it approaches atomic-scale discretization. However, in a finite element micromagnetic framework, energy terms are inherently dependent on the chosen cell size, which does not necessarily explain the offset field observed in Fig. 4(b). Moreover, the curvature dependence shown in Fig. 4(g) already captures the non-linear variation of the pinning field with curvature.

3) Based on SEM images of different nanostructures, the authors conclude that diameter modulation does not significantly contribute to DW pinning. However, Figures S7 and S9 suggest that DWs tend to become pinned in regions where the DW narrows, indicating a correlation between DW positioning and dips in thickness.

Overall, the BPDW appears to consistently stabilize in the straight regions of the nanowire, and the pinning field is higher for smaller radii of curvature, leading to a unidirectional ratchet-like motion. However, the proposed mechanism of DW pinning due to "curved regions of non-zero effective chirality" is not convincingly supported by the presented data. Therefore, I cannot recommend publication of the manuscript in its current form.

Reviewer #3

(Remarks to the Author)

I co-reviewed this manuscript with one of the reviewers who provided the listed reports. This is part of the Nature Communications initiative to facilitate training in peer review and to provide appropriate recognition for Early Career

Researchers who co-review manuscripts.

Version 1:

Reviewer comments:

Reviewer #2

(Remarks to the Author)

I appreciate the detailed responses and additional analyses the authors have provided, which have addressed much of the previous concerns and improved the manuscript. The revised version of the manuscript overall represents an interesting study on the influence of symmetry breaking on Bloch point textures through nano-curvature. It is potentially suitable for publication in Nature Communications, provided that the following concern can be adequately addressed on their claim regarding the pinning of the Bloch point by the atomic lattice:

“To model this pinning of the Bloch point singularity on the atomic lattice, we perform finite difference micromagnetic simulations of nanowires containing a BPDW, decreasing the cell size from the exchange length (4 nm) down to the atomic spacing (<1 nm). These simulations are equivalent to treating the exchange field as an atomistic Heisenberg exchange, where the Heisenberg exchange constant is scaled to match the macroscopic properties of the material. As a result, we expect the mesh pinning to correspond to a cell size matching the corresponding atomic lattice.”

This statement cannot be rigorously established using micromagnetic simulations, which is a continuum approach. Atomistic simulations are required to properly capture lattice-level pinning effects. While the pinning field does indeed vary with cell size, this is more likely a consequence of how micromagnetic energy terms scale with discretization, rather than evidence of true atomistic pinning. I would also expect similar behavior for other types of domain walls, not just BPDWs.

The constant offset observed across different nanostructures in the experiments is interesting. While atomistic pinning could be one contributing factor, the current work does not provide sufficient evidence to support that conclusion. This offset might arise from other factors, e.g. the internal structure of the BPDWs at this particular radius of curvature.

Reviewer #3

(Remarks to the Author)

We would like to begin by thanking the three referees for their comments. In the following, we address the comments of the referees one by one. The referee comments are given in bold, and our quotes from the manuscript are given in italics. Changes introduced to the paper and the supplementary materials are highlighted in red.

Reviewer #2 (Remarks to the Author):

Authors provide sufficient explanation on demand of the Referee. The revised manuscript is properly improved to deserve publication in Nature Communications.

We thank the referee for their positive assessment of our manuscript, and their recommendation that it be published in Nature Communications.

Reviewer #3 (Remarks to the Author):

In this study, domain wall (DW) motion in a curved 3D nanostructure was investigated. The authors have provided responses to the previous round of review comments. The microstructural characterization is particularly important for identifying the mechanisms underlying DW movement. The manuscript has been much improved. However, I still have the following concerns.

We appreciate the reviewer's recognition of the improvements made to the manuscript. We also thank the insightful questions, which have allowed us to further clarify our work. We address each of the raised points in detail below:

1) In many places of the manuscript, the effect of chirality was mentioned. For example, the authors state in Line 284 that they control local symmetry breaking and effective DMI by defining curved regions along the nanostructure. However, the observed pinning appears to result primarily from variations in magnetostatic energy due to changes in the radius of curvature, rather than from the chirality of the curvature. This is also evident from their micromagnetic simulations of the vortex domain wall.

In this work, we explore the influence of curvature-induced symmetry breaking on the stability and pinning of Bloch point domain walls. It is known that introducing curvature in magnetic systems can lead to an exchange-driven (or magnetostatics D. Sheka et al. Communications Physics volume 3, Article number: 128 (2020)) curvature-induced Dzyaloshinsky–Moriya-like interaction, effectively mimicking the behavior of a chiral unit cell in helimagnets and giving rise to a preferred chirality in the surrounding magnetic texture [Hertel, SPIN 03, 1340009 (2013), Sheka, APL 118, 230502 (2021), Sheka, Small 18, 2105219 (2022), Skoric et al., APL 118, 242403 (2021), Volkov et al., PRL 123, 077201 (2019)]. This mechanism has been described as a curvature-induced magnetochiral effect in recent literature and is relevant in the context of chiral textures such as transverse domain walls, and our observation that transverse walls preferentially form at the curved regions in our nanostructure.

We agree with the reviewer that for the Bloch point domain wall, the dominant mechanism for the observed domain wall pinning is the magnetostatic energy, and not the exchange-driven curvature-induced DMI. This is supported both by our micromagnetic simulations and experimental

measurements. To clarify this in the manuscript, we have taken care to refer to local symmetry breaking induced by the curvature, rather than curvature-induced chirality.

Nonetheless, our findings suggest that the energy of Bloch point-containing structures, such as domain walls, are modulated by changes in local symmetry. This insight into the interplay between symmetry and energy landscapes is highly relevant to systems such as chiral magnets, where symmetry breaking governs the stability and dynamics of topological textures.

2) The authors report that reducing the mesh size affects the pinning energy of the Bloch point domain wall (BPDW) as it approaches atomic-scale discretization. However, in a finite element micromagnetic framework, energy terms are inherently dependent on the chosen cell size, which does not necessarily explain the offset field observed in Fig. 4(b). Moreover, the curvature dependence shown in Fig. 4(g) already captures the non-linear variation of the pinning field with curvature.

We thank the referee for their comment, which we believe is referring to our finite difference simulations for the investigation of pinning of the Bloch point singularity on the atomic lattice. The referee is correct that the energetics of the Bloch point depend strongly on the mesh size. Here we make use of the property of finite difference simulations that the exchange field is equivalent to the atomistic Heisenberg exchange field predicted [Hubert and Schaefer, Magnetic Domains Chptr 2&5], meaning that by varying the cell size of the mesh, we are effectively varying the atomic lattice of the material, keeping the macroscopic properties of the system constant. As a result, when we perform our simulations as a function of mesh size, and observe the change in the depinning field, this is equivalent to performing atomistic simulations, and as such, we can conclude that the pinning is due to the pinning of the Bloch point on the atomic lattice, as previously. To clarify this point, we have updated the text as follows:

*“To model this pinning of the Bloch point singularity on the atomic lattice, we perform finite difference micromagnetic simulations of nanowires containing a BPDW, decreasing the cell size from the exchange length (4 nm) down to the atomic spacing (<1 nm). **These simulations are equivalent to treating the exchange field as an atomistic Heisenberg exchange, where the Heisenberg exchange constant is scaled to match the macroscopic properties of the material. As a result, we expect the mesh pinning to correspond to a cell size matching the corresponding atomic lattice. Indeed,** when we plot the depinning field as a function of cell size in Figure 4e, we observe a strong influence of the mesh on the pinning of the domain wall, which becomes non-negligible as the cell size approaches the atomic spacing determined with transmission electron microscopy (See Methods/ SI). **This indicates** that the deterministic/intrinsic pinning observed in the experiments could be related to the pinning of the Bloch point singularity on the atomic lattice within the nanostructures. This pinning of the Bloch point singularity on the atomic lattice is highly relevant not only for the dynamics of Bloch point domain walls, but also for other configurations and dynamic processes containing Bloch point singularities.”*

The referee is also correct that the curvature dependence of the depinning is given by the finite element simulations. Indeed, our approach was to separate these two contributions, to be able to address both individually. We have clarified this as follows:

“Having determined that the Bloch point singularity strongly influences the behaviour of the BPDW, we next consider the effect of curvature on the pinning of the texture. To determine the influence of

*curvature on this surrounding configuration – and to disentangle the effect of the central Bloch point singularity – we perform **separate finite element simulations** of a nanowire structure with the central 16 nm containing the singularity removed, effectively a thick-walled nanotube.”*

3) Based on SEM images of different nanostructures, the authors conclude that diameter modulation does not significantly contribute to DW pinning. However, Figures S7 and S9 suggest that DWs tend to become pinned in regions where the DW narrows, indicating a correlation between DW positioning and dips in thickness.

Overall, the BPDW appears to consistently stabilize in the straight regions of the nanowire, and the pinning field is higher for smaller radii of curvature, leading to a unidirectional ratchet-like motion. However, the proposed mechanism of DW pinning due to "curved regions of non-zero effective chirality" is not convincingly supported by the presented data. Therefore, I cannot recommend publication of the manuscript in its current form.

We thank the referee for their pertinent question. Indeed, there is some slight reduction in the thickness of the nanowire in the straight regions of the structure. However, when we consider the mechanism for the pinning, we see no correlation between the thickness change and the depinning field. On the contrary, we see a very strong correlation between the radius of curvature and the depinning field, providing strong evidence that the local curvature is responsible for the pinning of the Bloch point domain wall.

To address this question of the source of the pinning, and whether it is due to curvature or thickness, we conducted a statistical analysis of pinning locations across multiple nanostructures, detailed in Supplementary Figures S7–S9. Plotting the depinning field against diameter variation shows no clear correlation (Fig. S8b), whereas a linear dependence with curvature remains robust (Fig. S8a). Thus, although diameter variations may play a secondary role, our data indicate that curvature is the dominant mechanism.

We can quantitatively assess this correlation by calculating the correlation coefficient for the depinning field with respect to the thickness, and to the curvature. The correlation between the thickness and the depinning field is low, at [0.26], whereas the correlation between the curvature and the depinning field is much higher [-0.94] indicating that the curvature is the dominant effect for the pinning.

Figure S8. (a) Component of the magnetic field parallel to the wire as a function of the radius of curvature for four different nanostructures, where a clear linear trend can be observed. b) Component of the magnetic field parallel to the wire is plotted as a function of the variation in thickness for the same nanostructures, where no trend is observed.

We have updated the discussion in the SI as follows:

“By plotting the depinning field as a function of the radius of curvature and the change in diameter for the different nanostructures, there is a clear linear dependence of the depinning field on the radius of curvature but no clear trend related to the slope of geometric defects. Indeed, when we calculate the correlation coefficient between the depinning field and the thickness, and curvature, we obtain correlation coefficients of [0.26] and [-0.94], respectively. These coefficients indicate that there is a much stronger correlation between the curvature and the depinning field, than the thickness, providing evidence that the depinning is dominated by the local curvature. These results suggest again that while diameter changes can contribute to pinning, they are not the dominant mechanism here.”

Furthermore, it is important to note that if diameter variations were solely responsible for the observed domain wall positions, we would also expect transverse domain walls to be pinned in the same straight regions. However, in the same structure (Figure 2d), when a transverse domain wall is nucleated by applying a magnetic field perpendicular to the nanowire, the domain wall instead becomes pinned within the curved section. This behavior is attributed to the reduction in energy due to curvature effects.

In fact, for a transverse wall, the increase in wire thickness—such as that occurring in the curved region—leads to an increase in domain wall energy. This is because the larger domain wall size results in higher exchange and demagnetization energies. Therefore, if thickness variation were the dominant mechanism behind domain wall localization, the transverse wall would preferentially pin at thinner regions of the sample. In contrast, for a Bloch-point domain wall (BPDW), the energy change due to thickness variation is smaller. The curl structure of a BPDW does not contribute additional demagnetization energy (as it lacks surface charges), meaning that only the exchange contributes. Consequently, if the changes in the thickness doesn't dominate the pinning behavior of transverse domain walls, its influence is expected to be even lower for BPDWs.

These results support the conclusion that the pinning of domain walls—whether in straight or curved regions—is governed by the effect of curvature on the energy landscape of the specific domain wall

type. While we believe our data provides convincing evidence for the curvature-induced domain wall pinning, we agree with the referee that it is better to remove the mention of chirality. As a result, we now describe this phenomenon as: DW pinning due to “*curved regions of local symmetry breaking*”. We hope that with these changes to the manuscript, and clarifications of the wording, the referee can now support the publication of our manuscript in Nature Communications.

Reviewer #4 (Remarks to the Author):

We thank the referee.

We would like to begin by thanking the referees for their positive comments. In the following, we address the remaining comment of the referee. The referee comments are given in bold, and our quotes from the manuscript are given in italics. Changes introduced to the paper are highlighted in red.

Reviewer #2 (Remarks to the Author):

I appreciate the detailed responses and additional analyses the authors have provided, which have addressed much of the previous concerns and improved the manuscript. The revised version of the manuscript overall represents an interesting study on the influence of symmetry breaking on Bloch point textures through nano-curvature. It is potentially suitable for publication in Nature Communications, provided that the following concern can be adequately addressed on their claim regarding the pinning of the Bloch point by the atomic lattice:

“To model this pinning of the Bloch point singularity on the atomic lattice, we perform finite difference micromagnetic simulations of nanowires containing a BPDW, decreasing the cell size from the exchange length (4 nm) down to the atomic spacing (<1 nm). These simulations are equivalent to treating the exchange field as an atomistic Heisenberg exchange, where the Heisenberg exchange constant is scaled to match the macroscopic properties of the material. As a result, we expect the mesh pinning to correspond to a cell size matching the corresponding atomic lattice.”

This statement cannot be rigorously established using micromagnetic simulations, which is a continuum approach. Atomistic simulations are required to properly capture lattice-level pinning effects. While the pinning field does indeed vary with cell size, this is more likely a consequence of how micromagnetic energy terms scale with discretization, rather than evidence of true atomistic pinning. I would also expect similar behavior for other types of domain walls, not just BPDWs.

We thank the reviewer for pointing this out. We agree that standard micromagnetic theory is a continuum approximation and does not inherently resolve atomic-scale lattice effects. However, in finite-difference micromagnetics, the standard 3-point stencil used to compute the exchange field is mathematically equivalent to a classical Heisenberg model on a regular cuboid lattice. Consider the micromagnetic definition of the exchange field $\vec{H} = \frac{2A}{\mu_0 M_s} \cdot \nabla^2 \vec{m}$. For a 1D discretized finite-difference approximation this leads to the following expression for H

$$\vec{H}_i = \frac{2A}{\mu_0 M_s} \cdot \frac{\vec{m}_{i-1} - 2\vec{m}_i + \vec{m}_{i+1}}{dx^2}$$

with dx denoting the discretization length of the simulation cell. Computing the exchange-energy density from the discretized field as $e = -\frac{1}{2} \mu_0 M_s \vec{m} \cdot \vec{H}$ results in $e = -A \cdot \frac{\vec{m}_{i-1} \cdot \vec{m}_i + \vec{m}_i \cdot \vec{m}_{i+1} - 2\vec{m}_i \cdot \vec{m}_i}{dx^2}$. This expression exactly reproduces the Heisenberg exchange energy except for the constant -2 which can be neglected as constant contributions to the energy have no impact on the dynamics of the system. Generalizing to 3 dimensions and considering the cell volume $V = dx^3$ the micromagnetic exchange energy computed with a 3-point stencil is thus exactly equivalent to a next-neighbour Heisenberg coupling with $J = \frac{A \cdot V}{dx^2} = A \cdot dx$. By interpreting the discretization length dx as lattice constant a , the

micromagnetic simulation is thus equivalent to a simulation with atomistic exchange with the Heisenberg exchange integration J scaled according to $A \cdot dx$. We use this fact in order to investigate a potential lattice pinning. To clarify this point, we have revised the manuscript, which now reads::

" To model the pinning of the Bloch point singularity on the atomic lattice, we perform finite-difference micromagnetic simulations of nanowires containing a BPDW, systematically reducing the cell size from the exchange length (~ 4 nm) down to below the atomic lattice spacing (< 1 nm). In this context, we model the exchange interaction as an atomistic Heisenberg exchange, where atomic sites are placed at the centers of the simulation cells. The exchange integral is scaled with the lattice constant a as $J = A \cdot a$ with A being the exchange constant to ensure consistency with the macroscopic material parameters. This setup allows us to mimic the presence of the atomic lattice and investigate how it can act as a pinning potential for the Bloch point within a continuum micromagnetic framework. "

The constant offset observed across different nanostructures in the experiments is interesting. While atomistic pinning could be one contributing factor, the current work does not provide sufficient evidence to support that conclusion. This offset might arise from other factors, e.g. the internal structure of the BPDWs at this particular radius of curvature.

We agree with the referee that we cannot rule out contributions from other factors. Following the referee comment, we have revised the manuscript and added the following text:

*"This indicates that the deterministic/intrinsic pinning observed in the experiments could be related to the pinning of the Bloch point singularity on the atomic lattice within the nanostructures **although the contribution of other factors related to the geometry and Bloch point spin structure cannot be fully discarded.**"*

Reviewer #3 (Remarks to the Author):

We thank the referee.